# Generalized Delayed Feedback Model
# with Post-Click Information in Recommender Systems

**Jia-Qi Yang**    **De-Chuan Zhan**[*]
State Key Laboratory for Novel Software Technology
Nanjing University, Nanjing, 210023, China
yangjq@lamda.nju.edu.cn, zhandc@nju.edu.cn

## Abstract

Predicting conversion rate (e.g., the probability that a user will purchase an item) is a fundamental problem in machine learning based recommender systems. However, accurate conversion labels are revealed after a long delay, which harms the timeliness of recommender systems. Previous literature concentrates on utilizing early conversions to mitigate such a delayed feedback problem. In this paper, we show that post-click user behaviors are also informative to conversion rate prediction and can be used to improve timeliness. We propose a generalized delayed feedback model (GDFM) that unifies both post-click behaviors and early conversions as stochastic post-click information, which could be utilized to train GDFM in a streaming manner efficiently. Based on GDFM, we further establish a novel perspective that the performance gap introduced by delayed feedback can be attributed to a temporal gap and a sampling gap. Inspired by our analysis, we propose to measure the quality of post-click information with a combination of temporal distance and sample complexity. The training objective is re-weighted accordingly to highlight informative and timely signals. We validate our analysis on public datasets, and experimental performance confirms the effectiveness of our method.

## 1 Introduction

Conversion rate (CVR) prediction has become a core problem in display advertising with the prevalence of the cost-per-conversion (CPA) payment model[1–3]. With CPA, advertisers bid for predefined user behaviors such as purchases or downloads. Similar demands also arise from online retailers that aim to improve sales volume[4], which need to recommend items with high conversion rates to users. However, conversions (e.g., purchases) may happen after a long delay [5], which leads to a delay in conversion labels. The adverse impact of such a delayed feedback problem becomes increasingly significant with a rapidly changing market, where the features of users and items are updated every second. Thus, how to timely update the CVR prediction model with delayed feedback attracts much attention in recent years [5–11].

Since conversions are gradually revealed after click events, previous literature concentrates on utilizing available conversion labels that reveal earlier. Inserting early conversions into the training data stream will also introduce many fake negative samples [6], which will eventually convert but have not converted yet. To mitigate the adverse impact of fake negatives, Chapelle [5] proposes a delayed feedback model (DFM) to model conversion rate along with expected conversion delay, then the model is trained to maximize the likelihood of observed labels. The delayed feedback model is further improved by introducing kernel distribution estimation method [12] and more expressive neural networks [10, 13, 14]. However, the need to maintain a long time scale offline dataset impedes

---

[*]De-Chuan Zhan is the corresponding author.

36th Conference on Neural Information Processing Systems (NeurIPS 2022).

training efficiency of DFM based methods in large scale recommender systems[6]. To enable efficient training in a streaming manner, Ktena et al. [6] proposes to insert a negative sample once a click sample arrives and re-insert a duplicated positive sample once its conversion is observed. The fake negative samples are supposed to be corrected by an importance sampling[15] method. Importance sampling based methods are further improved by adopting different sampling strategies [7–9, 16].

Besides the conversion labels, many events related with conversion exist in real-world recommender systems[17–19]. For example, after clicking through an item, a user may decide to add this item to a shopping cart. Statistical data reveals that such post-click behaviors have a strong relationship with conversions: About 12% of items in a shopping cart will finally be bought, while the proportion is less than 2% without entering a shopping cart[17]. Intuitively, utilizing post-click behaviors can potentially mitigate the delayed feedback problem in CVR prediction since the time delay of post-click behaviors is usually much shorter than conversions. However, the corelations between post-click behaviors and conversions are far more complex than only considering conversions, and the problem is further complicated by the streaming nature of training with delayed feedback.

In this work, we propose a Generalized Delayed Feedback Model (GDFM) which generalizes the delayed feedback model (DFM)[5] to unify post-click information and early conversions. Based on GDFM, we establish a novel view on learning with delayed feedback. We argue that training using post-click information in the delayed feedback problem is intrinsically different from traditional learning problems from three perspectives: (**i**) Estimating conversion rates via post-click actions requires more samples than using conversion labels directly, which highlights the importance of sample complexity. (**ii**) The post-click actions bring information of past distributions, which incurs a temporal gap. (**iii**) The signals provided by post-click actions are highly stochastic and lead to large variance on the target model during training, which may instead hinder the performance of the conversion rate prediction. Inspired by our analysis, we propose to (**i**) measure the information carried by a post-click action with conditional entropy, which we empirically show is related to the sample complexity of estimating the conversion rate; (**ii**) measure the temporal distribution gap by time delay; (**iii**) stabilize streaming training with a regularizer trained on past data distribution. We conduct various experiments on public datasets with delayed feedback under a streaming training and evaluation protocol. Experimental results validate the effectiveness of our method. To summarize,

- We propose a generalized delayed feedback model (GDFM) to support various user behaviors beyond conversion labels and arbitrary revealing times of user actions.

- From a novel perspective, we attribute the difficulty of efficiently utilizing user behaviors to a sampling gap and a temporal gap in delayed feedback problems.

- Based on our analysis, we propose a re-weighting method that could selectively use the user actions that are most informative to improve performance. Our method achieves stable improvements compared to baselines.

## 2 Background

In recommender systems, the data stream is collected from user behavior sequence. Without loss of generality, we take an E-Commerce search engine as an example, while a similar procedure also holds for display advertising[2, 3] and personalized recommender[4]. When a user searches for a keyword, the search engine will provide several items for this user. Here, a (user-keyword-item) tuple is called an *impression*, which corresponds to a sample $\mathbf{x}$. After viewing an item's detail page, the user may purchase it. The probability that a user will buy after clicking is called conversion rate (CVR). Conversion can be any desired user behavior, such as registering an account or downloading a game. Without loss of generality, we only consider one type of conversion in the rest of this paper, while our method can be applied to multi-class case[20] without modification. If an impression sample $\mathbf{x}$ is finally converted, it will be labeled as $\mathbf{y} = 1$ and $\mathbf{y} = 0$ otherwise. The conversion rate of $\mathbf{x}$ corresponds to $p(\mathbf{y} = 1|\mathbf{x})$.

The distribution $p(\mathbf{x})$ and $p(\mathbf{y}|\mathbf{x})$ changes rapidly in real-world recommender systems. For example, when a promotion starts, the conversion rate may increase steeply for items with a large discount, and such promotion happens every day. The CVR model has to be updated timely to capture such distribution change. However, the ground-truth conversion labels are available only after a long delay: Many users purchase several days after a click event $\mathbf{x}$[5]. Usually, a sample will be

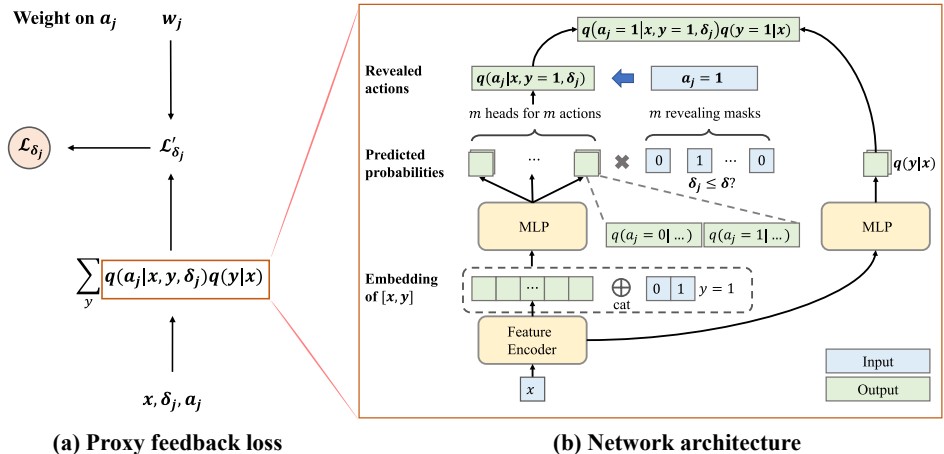

**(a) Proxy feedback loss**    **(b) Network architecture**

Figure 1: (a) The proxy feedback loss on the $j$th action $\mathbf{a}_j$. (b) The neural network architecture of GDFM.

labeled as negative after a long enough waiting time, e.g., 30 days[5], this delay time of conversion label is denoted as $\delta_y$. Besides conversion labels, post-click actions (behaviors) are denoted as $\mathbf{a}_j, j \in \{1, ..., m\}$. Each action $\mathbf{a}_j$ is paired with a revealing time $\delta_j$, which means the value of $\mathbf{a}_j$ is determined after a $\delta_j$ delay. For example, we can query the database to see whether a user has put an item into the shopping cart 10 minutes ($\delta_j$) after clicking. If true, this action is labeled as $\mathbf{a}_j = 1$, and $\mathbf{a}_j = 0$ otherwise. There can be several revealing times for a single type of action, e.g., we may choose to reveal the shopping cart information at 10 minutes, 30 minutes and 1 hour, they are treated as different actions since the revealing time is different. It is noteworthy that revealing the conversion labels is also treated as a type of action. To capture the characteristic that the data distribution changes along with time, the data distribution is denoted by $p^t(\mathbf{x}, \mathbf{y}) = p^t(\mathbf{x})p^t(\mathbf{y}|\mathbf{x})$ at time $t$. Our goal is to predict $p^t(\mathbf{y}|\mathbf{x})$ at time $t$ by training a model $q_\theta(\mathbf{y}|\mathbf{x})$, where $\theta$ denotes model parameters. Without data distribution shift, it is straightforward to optimize likelihood by sampling from $p^t$. However, in the delayed feedback problem, the latest available labels have a $\delta_y$ delay. Thus, simply training using available labeled data will lead to $q_\theta(\mathbf{y}|\mathbf{x}) = p^{t-\delta_y}(\mathbf{y}|\mathbf{x})$ instead.

## 3 Generalized Delayed Feedback Model

We propose the following generalized delayed feedback model to describe the relationship between data features $\mathbf{x}$, conversion labels $\mathbf{y}$, post-click actions $\mathbf{a}$, revealing time of actions $\delta$, and data distribution at time $t$:

$$p^t(\mathbf{x}, \mathbf{y}, \mathbf{a}, \delta) = p^t(\mathbf{y}|\mathbf{x})p(\mathbf{a}|\mathbf{x}, \mathbf{y}, \delta)p(\delta)p^t(\mathbf{x}) \tag{1}$$

Our formulation unified previous delayed feedback models[5, 10–12] while enable more general feedback and revealing times. $p^t(\mathbf{y}|\mathbf{x})$ is the target CVR distribution, we assume that $p^t(\mathbf{y}|\mathbf{x})$ is independent to user actions $\mathbf{a}$, which enables us to extend the existing CVR prediction framework seamlessly. To model the post-actions, we introduce a post-action distribution $p(\mathbf{a}|\mathbf{x}, \mathbf{y}, \delta)$, which depends on sample features $\mathbf{x}$, the conversion label $\mathbf{y}$, and the revealing time $\delta$. We assume the post-action distribution $p(\mathbf{a}|\mathbf{x}, \mathbf{y}, \delta)$ is fixed or changes much slower than $p^t(\mathbf{y}|\mathbf{x})$, so that the post-action distribution does not depend on time $t$. This assumption is critical for learning under delayed feed-back since the relationship between post-actions and conversion should be stable so that post-actions can be informative. Previous literature also relies on similar assumptions implicitly, for example, a predictable delay time of conversion[5] or a loss function that depends on a stable post-action distribution in our formulation[7–9]. Our formulation decouples the post-action distribution and the conversion distribution, which enables us to formulate this assumption explicitly and conduct further analysis based on this assumption. An alternative but more intuitive definition is to define $p^t(\mathbf{x}, \mathbf{y}, \mathbf{a}, \delta) = p^t(\mathbf{a}|\mathbf{x}, \delta)p(\mathbf{y}|\mathbf{x}, \mathbf{a}, \delta)p(\delta)p(\mathbf{x})$, which requires to sum out $\mathbf{a}$ and $\delta$ to make a prediction. We discuss this approach in the supplementary material. We further introduce a revealing

time distribution of post-actions $p(\delta)$, which is independent of other variables. $\delta$ is also known as the elapsed time in previous literature [5, 8]. The revealing time enables us to inject action information at controllable time $\delta$, which is typically less than $\delta_y$ so that the model can be updated much earlier. We introduce a streaming training method of GDFM in the next section.

## 3.1 Training GDFM

We introduce an action prediction model $q_\phi(\mathbf{a}|\mathbf{x}, \mathbf{y}, \delta)$ to estimate the action distribution $p(\mathbf{a}|\mathbf{x}, \mathbf{y}, \delta)$, the CVR prediction model is denoted as $q_\theta(\mathbf{y}|\mathbf{x})$, which will be used to predict the conversion rate $p^t(\mathbf{y}|\mathbf{x})$. In the data stream, not all the information in GDFM defined in Eq. (1) is available at that same time, so we propose a streaming training method for GDFM that can utilize available information in the data stream. According to the availability of $\mathbf{a}, \mathbf{y}$, there are several different cases. We assume the current timestamp is $t$.

**When $\mathbf{a}, \mathbf{x}, \mathbf{y}$ is available**  The action prediction model $q_\phi(\mathbf{a}|\mathbf{x}, \mathbf{y}, \delta)$ can be trained with following loss when $\mathbf{a}, \mathbf{x}$ and $\mathbf{y}$ are available.

$$\mathcal{L}_{\text{action}} = -\mathbb{E}_{p^{t-\delta_y}} \log q_\phi(\mathbf{a}_j|\mathbf{x}, \mathbf{y}, \delta_j) \tag{2}$$

Note that with the assumption that $p(\mathbf{a}_j|\mathbf{x}, \mathbf{y}, \delta_j)$ is invariant with respect to $t$, loss Eq. (2) will be minimized when $p(\mathbf{a}_j|\mathbf{x}, \mathbf{y}, \delta_j) = q_\phi(\mathbf{a}_j|\mathbf{x}, \mathbf{y}, \delta_j)$.

**When $\mathbf{y}, \mathbf{x}$ is available**  When we have the ground-truth label of $\mathbf{y}$, we are able to update the CVR model directly by minimizing cross entropy.

$$\mathcal{L}_{\delta_y} = -\mathbb{E}_{p^{t-\delta_y}} \log q_{\theta^{t-\delta_y}}(\mathbf{y}|\mathbf{x}) \tag{3}$$

Note that this loss is minimized when $p^{t-\delta_y}(\mathbf{y}|\mathbf{x}) = q_{\theta^{t-\delta_y}}(\mathbf{y}|\mathbf{x})$.

**When $\mathbf{a}, \mathbf{x}$ is available**  Since the delay times of user actions ($\delta_j$) are smaller than the conversion label delay $\delta_y$, we can update the CVR model once a user action is observed. When we observe the $j$th user action $\mathbf{a}_j$ at revealing time $\delta_j$, we can update the prediction model $q_\theta(\mathbf{y}|\mathbf{x})$ using following *proxy feedback loss*:

$$\mathcal{L}'_{\delta_j} = -\mathbb{E}_{p^{t-\delta_j}} \log q(\mathbf{a}_j|\mathbf{x}, \delta_j) = -\mathbb{E}_{p^{t-\delta_j}} \log \sum_{\mathbf{y}} q_\phi(\mathbf{a}_j|\mathbf{x}, \mathbf{y}, \delta_j) q_\theta(\mathbf{y}|\mathbf{x}) \tag{4}$$

which is depicted in Figure. 1 (a). However, it is unclear whether minimizing Eq. (4) will help learning $p^t(\mathbf{x}|\mathbf{y})$.

## 3.2 Narrowing the delayed feedback gap with post-actions

The training objective of the proxy feedback loss can be viewed as a multi-task training problem with different actions, which is also explored by Hou et al. [10] and Li et al. [11]. However, learning with delayed feedback is different from general multi-task learning problems without delayed feedback. In learning with delayed feedback, since we are training on different data distribution at different times, the tasks are intrinsically conflicted, which may be harmful to the target [10, 21]. We analyze its effect in the following sections.

### 3.2.1 Post-actions can be informative

Since in GDFM we are using post-actions from $t - \delta_j$ instead of $t - \delta_y$, we have a chance to do better. But will optimizing Eq. (4) improve the performance of $q(\mathbf{y}|\mathbf{x})$? Without loss of generality, we consider an action $\mathbf{a} \in \{a_1, ..., a_k\}$, and $p(\mathbf{a}|\mathbf{x}) = \sum_{\mathbf{y}} p(\mathbf{y}|\mathbf{x}) p(\mathbf{a}|\mathbf{x}, \mathbf{y})$ (condition on $\delta$ omitted), $\mathbf{y} \in \{y_1, ..., y_n\}$. Assuming we can estimate $p(\mathbf{a}|\mathbf{x})$ accurately and $k \geq n$, we have following results:

**Lemma 3.1.** *Use a matrix $\mathbf{M_x} \in \mathbb{R}^{k \times n}$ to denote the conditional probability $p(\mathbf{a}|\mathbf{x}, \mathbf{y})$, where $(\mathbf{M_x})_{ji} = p(\mathbf{a} = a_j|\mathbf{x}, \mathbf{y} = i)$. We can recover $p(\mathbf{y}|\mathbf{x})$ from $p(\mathbf{a}|\mathbf{x})$ if and only if $\mathrm{rank}(\mathbf{M_x}) = n$.*

**Proof sketch**: $p(\mathbf{a}|\mathbf{x})$ is a linear transformation from $p(\mathbf{y}|\mathbf{x})$ and the transform matrix is given by $\mathbf{M_x}$. By construction, a solution exists and $\mathrm{rank}(\mathbf{M_x}) = n$ guarantees that the solution is unique.

Lemma. 3.1 highlights the benefits of utilizing post-actions: if the relationship between a post-action and the target is predictable (we can train a model $q(\mathbf{a}|\mathbf{x}, \mathbf{y})$ to approximate $p(\mathbf{a}|\mathbf{x}, \mathbf{y})$) and informative ($\text{rank}(M) = n$) we can recover the target distribution even without the ground-truth label $\mathbf{y}$. Since $\text{rank}(\mathbf{M}) < n$ requires that some rows of $\mathbf{M}$ are linear combination of the others, which rarely happens in real-world problems, we assume the assumption holds true in the next section. We discuss the case that this assumption fails in Section. 3.2.4. However, even if $\text{rank}(\mathbf{M}) = n$, recovering $p(\mathbf{y}|\mathbf{x})$ is still challenging. Because we are estimating $p(\mathbf{a}|\mathbf{x})$ using samples from $p(\mathbf{a}|\mathbf{x})$, and the difficulty of estimating $p(\mathbf{y}|\mathbf{x})$ via estimating $p(\mathbf{a}|\mathbf{x})$ also depends on properties of $p(\mathbf{a}|\mathbf{x}, \mathbf{y})$.

### 3.2.2 Measuring information carried by actions

Not all actions carry the same amount of information about $p(\mathbf{y}|\mathbf{x})$. For example, an action may carry zero information about $\mathbf{y}$ if it is constant and irrelevant to $\mathbf{y}$; it may carry full information if it always equals $\mathbf{y}$. The general relationship between $\mathbf{a}$ and $\mathbf{y}$ is far more complex, which lies between non-informative and fully informative. We want to estimate the information quantity of each action to utilize actions more efficiently. To this end, we propose to use conditional entropy to measure information carried by actions. The conditional entropy of $\mathbf{y}$ given $\mathbf{a}$ can by calculated by

$$H(\mathbf{y}|\mathbf{a}) = \sum_{\mathbf{a},\mathbf{y}} p(\mathbf{y}, \mathbf{a}) \log \frac{p(\mathbf{a})}{p(\mathbf{y}, \mathbf{a})} \tag{5}$$

Conditional entropy measures the amount of information needed to describe $\mathbf{y}$ given the value of $\mathbf{a}$. When $\mathbf{a}$ is non-informative to $\mathbf{y}$, that is $p(\mathbf{a}, \mathbf{y}) = p(\mathbf{a})p(\mathbf{y})$, we have $H(\mathbf{y}|\mathbf{a}) = H(\mathbf{y})$, which is the maximum value of $H(\mathbf{y}|\mathbf{a})$; when $\mathbf{a}$ determines value of $\mathbf{y}$, that is, there is a function $\mathbf{y} = f(\mathbf{a})$, we have $H(\mathbf{y}|\mathbf{a}) = 0$. Since the scale of $H(\mathbf{y}|\mathbf{a})$ only depends on $p(\mathbf{y})$, this information metric is comparable among different $\mathbf{a}$, which is a desired property.

Empirically, we found that conditional entropy is also related to the sample complexity of estimating $p(\mathbf{y}|\mathbf{x})$ via $p(\mathbf{a}|\mathbf{x}) = \sum_{\mathbf{y}} p(\mathbf{a}|\mathbf{x}, \mathbf{y})p(\mathbf{y}|\mathbf{x})$ ($\delta$ omitted) as in Eq. (4). In learning with delayed feedback, we are sampling from different distribution $p^t$ at different time stamp $t$, which indicates that to perform well, we should be able to learn fast. Thus, analysis that is based on a large amount of i.i.d samples from a fixed distribution can't capture the difficulty of learning with delayed feedback. Property testing [22] considers sample complexity of learning distributions. Specifically, distinguishing between two discrete distributions $p$ and $q$ with a probability at least $1 - \epsilon$ requires $\Omega(\frac{1}{D_{\text{TV}}^2(p,q)} \log \frac{1}{\epsilon})$ samples from $q$ and $p$.

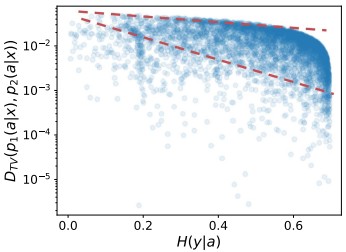

Figure 2: Conditional entropy and transformed distance.

Where $D_{TV}$ is the total variance distance which is defined by $D_{\text{TV}}(p, q) = \sum_i |p(i) - q(i)|$. Since we are estimating $p(\mathbf{y}|\mathbf{x})$ through a proxy distribution $p(\mathbf{a}|\mathbf{y})$, the sample complexity depends on how close the transformed distribution $p(\mathbf{a}|\mathbf{x})$ is. Empirically, we found that using $p(\mathbf{a}|\mathbf{x})$ will make $p(\mathbf{y}|\mathbf{x})$ more difficult to learn, that is, the distance of two distribution $p_1(\mathbf{y}|\mathbf{x})$ and $p_2(\mathbf{y}|\mathbf{x})$ will decrease when they are transformed to $p_1(\mathbf{a}|\mathbf{x})$ and $p_2(\mathbf{a}|\mathbf{x})$ correspondingly. So the difficulty of learning $p(\mathbf{y}|\mathbf{x})$ via $p(\mathbf{a}|\mathbf{x})$ depends on the change of distribution distance incurred by the stochastic transformation $p(\mathbf{a}|\mathbf{y})$. We empirically investigate the relationship between the change of distribution distance and conditional entropy by Monte Carlo sampling, the results in Figure. 2 indicate that as the conditional entropy increases, the transformed distribution distance decreases exponentially, which motivates the following information weights

$$w_{\text{info}}(p(\mathbf{a}_j, \mathbf{y})) = e^{-\alpha H(\mathbf{y}|\mathbf{a}_j)} \tag{6}$$

where $\alpha$ is a hyper-parameter. Eq. (6) roughly measures the reciprocal of sample complexity of learning $p^{t-\delta_j}(\mathbf{y}|\mathbf{x})$ via $p^{t-\delta_j}(\mathbf{a}_j|\mathbf{x})$, so that actions are weighted based on their effective information carried by each sample.

### 3.2.3 Measuring temporal gap

Besides the sample complexity of estimating $p^{t-\delta_j}(\mathbf{y}|\mathbf{x})$ via action distribution, the intrinsic temporal gap between $p^t$ and $p^{t-\delta_j}$ also influence the predicting performance. Even if we have unlimited

samples from $p^{t-\delta_j}$ we are only able to recover $p^{t-\delta_j}(\mathbf{y}|\mathbf{x})$ instead of $p^t(\mathbf{y}|\mathbf{x})$. So we also need to take the revealing time $\delta_j$ into consideration. However, we can't estimate the gap between $p^{t-\delta_j}$ and $p^t$ accurately since the exact distribution is unknown and changes along with time. Empirically (Figure. 3) we found that the gap measured by KL divergence tends to increase as the time gap increases. So we propose the following temporal weight

$$w_{\text{time}}(\delta_j) = e^{-\beta \delta_j} \tag{7}$$

which measures the gap between $p^t$ and $p^{t-\delta_j}$. $\beta$ is a hyper-parameter that reflects the expected changing speed of data distribution along with time, larger value of $\beta$ indicates that we discard more information from stale data.

Overall, we introduce a weight on loss Eq. (4) as follows:

$$\mathbf{w}_j = w(p(\mathbf{a}_j, \mathbf{y}), \delta_j) = w_{\text{info}}(p(\mathbf{a}_j, \mathbf{y})) w_{\text{time}}(\delta_j) \tag{8}$$

For simplicity, our $\mathbf{w}_j$ does not depend on $\mathbf{x}$, the extension to $\mathbf{x}$ dependent weights is straightforward. The weight $\mathbf{w}_j$ can be decomposed into two components, the first component $w_{\text{info}}$ measures how informative action $\mathbf{a}_j$ is to the target $\mathbf{y}$. The second component $w_{\text{time}}$ measures the gap between $p^t$ and $p^{t-\delta_j}$. The procedure of calculating weight vector $\mathbf{w} \in \mathbb{R}^m$ is summarized in Algorithm. 2.

| **Algorithm 1** Streaming training of GDFM | **Algorithm 2** Estimating $w_j$ |
|---|---|
| 1: **for** $\{\mathbf{x}, \mathbf{y}, \mathbf{a}, \delta\}$ in data stream **do** | **Input**: Dataset $\{\mathbf{y}, \mathbf{a}\}$ |
| 2:    **for** $\{\mathbf{x}, \mathbf{y}, \mathbf{a}\}$ that $\mathbf{y}$ is unlabeled **do** | **Parameter**: $\alpha, \beta$. |
| 3:       Update $q_\theta(\mathbf{y}|\mathbf{x})$ with Eq. (10). | |
| 4:    **end for** | 1: Estimate $p(\mathbf{a}, \mathbf{y})$ by counting. |
| 5:    **for** $\{\mathbf{x}, \mathbf{y}, \mathbf{a}\}$ that $\mathbf{y}$ is labeled **do** | 2: Calculate $H(\mathbf{y}|\mathbf{a})$ with Eq. (5). |
| 6:       Update $q_\phi(\mathbf{a}|\mathbf{x}, \mathbf{y}, \delta)$ with Eq. (2). | 3: Calculate $w_{\text{info}}$ with Eq. (6). |
| 7:       Update $q_{\theta^{t-\delta_y}}(\mathbf{y}|\mathbf{x})$ with Eq. (3). | 4: Calculate $w_{\text{time}}$ with Eq. (7). |
| 8:    **end for** | 5: Calculate weights $\mathbf{w}_j$ with Eq. (8). |
| 9: **end for** | 6: Normalize $\mathbf{w}$ to have $\text{mean}(\mathbf{w}) = 1$. |

### 3.2.4   Reducing variance by delayed regularizer

Estimating probability with sampling will incur instable variance during stream training, which may harm performance. Since we are estimating using a stochastic proxy $p(\mathbf{a}|\mathbf{x})$, the variance will be even higher. To reduce variance and stabilize training, we introduce a regularizer loss Eq. (9) that constraints update step during training, where $q_{\theta^{t-\delta_y}}(\mathbf{y}|\mathbf{x})$ is trained with loss Eq. (3).

$$\mathcal{L}_{\text{reg}} = \text{KL}(q_{\theta^{t-\delta_y}}(\mathbf{y}|\mathbf{x}) || q_\theta(\mathbf{y}|\mathbf{x})) \tag{9}$$

Besides reducing variance during training, another critical function of $\mathcal{L}_{reg}$ is to make GDFM *safe* to introduce more actions. The analysis in Section. 3.2.1 shows that if an action is informative, we can learn $p(\mathbf{y}|\mathbf{x})$ through $p(\mathbf{a}|\mathbf{x})$. However, if $\text{rank}(\mathbf{M}) < n$ (typically because $k < n$) or the information carried by $p(\mathbf{a}|\mathbf{x})$ is too weak to recover $p(\mathbf{y}|\mathbf{x})$ as analyzed in Section. 3.2.2, GDFM may fail to learn $p(\mathbf{y}|\mathbf{x})$ because of interference from action signals. The regularizer loss $\mathcal{L}_{reg}$ introduces the ground-truth labels of $\mathbf{y}$ from $p^{t-\delta_y}$, which guarantees that the performance learned by GDFM will not be much worse than $p^{t-\delta_y}$ (the Vanilla method in experiments). In this perspective, $\mathcal{L}_{reg}$ make it safer to introduce various post-click information into our training pipeline without worrying about sudden performance drop, which is critical in production environments. The overall loss function with revealing time $\delta_j$ is

$$\mathcal{L}_{\delta_j} = \mathbf{w}_j \mathcal{L}'_{\delta_j} + \lambda \mathcal{L}_{\text{reg}} \tag{10}$$

The streaming training procedure of GDFM is summarized in Algorithm. 1.

## 4   Experiments

### 4.1   Datasets and data analysis

We use two large-scale real-world datasets: 1) **Criteo Conversion Logs**[2] is collected from an online display advertising service within 60 days and consists of about 16 million samples with conversion

---

[2]`https://labs.criteo.com/2013/12/conversion-logs-dataset/`

labels and timestamps[5]. 2) **Taobao User Behavior**[3] is a subset of user behaviors on Taobao collected within 9 days and consists of more than 70 million samples and 1 million users[23]. Taobao dataset provides user behaviors within (page-view, buy, cart, favorite).

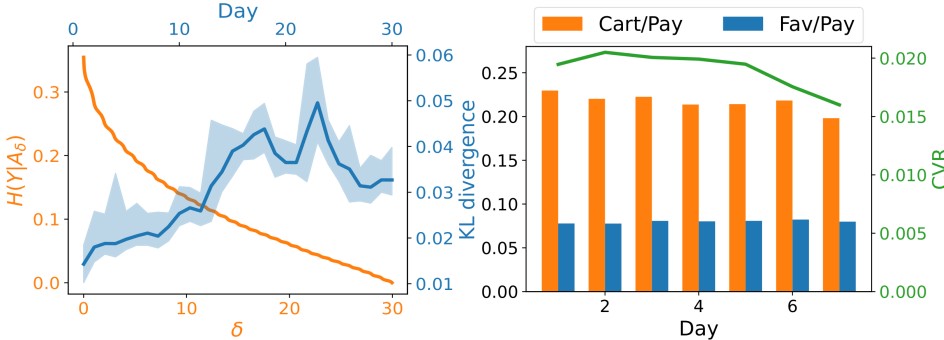

Figure 3: (Left) Conditional entropy (orange) and temporal gap (blue) changes along with time in Criteo dataset. (Right) In Taobao dataset, the conditional distribution $p(\mathbf{a}|\mathbf{y})$ (bar) is stabler than conversion rate (line).

We conduct data analysis on the datasets to validate our assumption on GDFM. The results are depicted in Figure. 3. First, we investigate the change of conversion rate distribution $p(\mathbf{y}|\mathbf{x})$. Since exact $p(\mathbf{y}|\mathbf{x})$ is not available, we train a CVR model on the first 30 days to estimate the conversion rate at the 30th day, which is denoted as $p^0(\mathbf{y}|\mathbf{x})$. Then we fine tune this model day by day to estimate the conversion rate at $p^t(\mathbf{y}|\mathbf{x})$, here $t$ denotes the $t$th day. In this way, we can estimate the temporal gap between $p^0(\mathbf{y}|\mathbf{x})$ and $p^t(\mathbf{y}|\mathbf{x})$ by their Kullback-Leibler divergence $D_{KL}(p^0||p^t)$. In Figure. 3 (Left) we can see that the temporal gap grows along with time, which necessitate the analysis of time dependent conversion distribution $p^t(\mathbf{y}|\mathbf{x})$ in Section. 3. Secondly, we investigate the information quality of actions with conditional entropy as proposed in Section. 3.2.2. Specifically, we use the observed conversion at revealing time $\delta$ as post-action $A_\delta$, and vary the revealing time from 0 to 30 days. We can infer from Figure. 3 (Left) that the information carried by this action steadily increase (the conditional entropy decrease) along with time, which is intuitive that the latter actions are more informative. Thirdly, we investigate the relationship between post-click actions and conversions. We plot the conversion rate and the proportion of cart and favorite actions in converted samples (which corresponds to $p(\mathbf{a} = 1|\mathbf{y} = 1)$) in the Taobao dataset in Figure. 3 (Right). We can see that the action distribution $p(\mathbf{a} = 1|\mathbf{y} = 1)$ is stabler comparing to conversion rate, which conforms our assumption of stable action distribution in Section. 3.

## 4.2 Evaluation

**Streaming evaluation** [4] We follow a streaming evaluation protocol proposed by Yang et al. [8]. Specifically, the datasets are split into pretraining and streaming datasets. An initial conversion rate prediction model as well as auxiliary models (e.g., the importance weighting model in ESDFM and action distribution model $q_\phi$ in GDFM) are trained on this pretraining dataset to simulate a stable state after a long time of streaming training in real-world recommender systems. Then the models are evaluated and updated hour by hour in the streaming dataset. Each method is trained with the available information at the corresponding timestamps. Following practice in [8–10, 16], we report the receiver operating characteristic curve (AUC), precision-recall curve (PR-AUC) and negative log-likelihood (NLL). The metrics are calculated within each hour. We report the average performance on the streaming dataset.

**Baseline methods** To evaluate the performance of GDFM, we compare with the following methods: 1) **Pretrain**: The CVR model is trained on the pretraining dataset, then fixed in streaming evaluation. The following methods start with this pre-trained model at the beginning of the streaming evaluation. 2) **Vanilla**: Wait for $\delta_y$ time, then use conversion labels to train the model. 3) **Oracle**: Use conversion labels to train without delay. The oracle method corresponds to the upper bound of performance

---

[3]https://tianchi.aliyun.com/dataset/dataDetail?dataId=649&userId=1&lang=en-us
[4]code available at https://github.com/ThyrixYang/gdfm_nips22

with feedback delay. We compare two representative importance sampling methods. 4) **FNW**[6]: A sample is labeled as negative on arrival, and a duplicate is inserted once its conversion is observed. The fake negative weighted (FNW) loss adjusts the loss function; 5) **ES-DFM**[8]: After waiting for a predefined time, a sample is labeled as negative if conversion has not been observed. If a sample converts after the waiting time, a duplicate is inserted. The loss function is adjusted by the ES-DFM loss. We also compare with a multi-task learning method based on DFM. 6) **MM-DFM**[10]: Treating predicting the observed conversion label as a multi-task learning problem, the tasks are optimized jointly using streaming data.

Table 1: Performance of compared methods on Criteo and Taobao dataset. The Pretrain method corresponds to 0%, and the Oracle method corresponds to 100%, their absolute performance is in parentheses. We also report the standard deviation of each method with 5 different random seeds.

| Method | Criteo | | | Taobao | | |
|---|---|---|---|---|---|---|
| | AUC | PR-AUC | NLL | AUC | PR-AUC | NLL |
| Pretrain | 0.0%(0.815) | 0.0%(0.607) | 0.0%(0.414) | 0.0%(0.703) | 0.0%(0.054) | 0.0%(0.084) |
| Vanilla | 25.2±0.2% | 25.1±0.2% | 24.6±0.1% | 58.9±0.8% | 61.7±1.7% | 46.0±1.9% |
| FNW[6] | 62.0±0.3% | 43.1±0.5% | 40.3±1.2% | 39.6±0.8% | -30.5±1.7% | -361±12% |
| ES-DFM[8] | 71.4±0.5% | 63.3±1.1% | 66.2±1.2% | 62.6±0.7% | 19.6±3.2% | -214±1.2% |
| MM-DFM[10] | 69.7±1.2% | 39.2±6.7% | 54.2±3.6% | 59.8±3.6% | 62.1±2.6% | -14.9±10.5% |
| GDFM(ours) | **74.9±0.7%** | **68.1±1.6%** | **72.4±0.6%** | **79.4±0.5%** | **80.7±0.9%** | **49.6±3.1%** |
| Oracle | 100%(0.841) | 100%(0.642) | 100%(0.389) | 100%(0.724) | 100%(0.063) | 100%(0.083) |

**Implementation**  Following [8, 9], we discretize and treat numerical features the same as categorical features in the Criteo dataset. Since the Taobao dataset does not provide user features, we use the last 5 user behaviors as features[23]. Inspired by Weinberger et al. [24], we hash user ID and item ID into bins and use an embedding to represent each bin. We use the same architecture for all the methods to ensure a fair comparison. All the methods are carefully tuned. We use $\alpha = 2$, $\beta = 1$, $\lambda = 0.01$, $lr = 10^{-3}$ for GDFM. The network structure and procedure to calculate the proxy feedback loss Eq. (4) used by GDFM is depicted in Figure. 1 (b).

The performance is reported in Table. (1). Following [8, 9, 16], we report the relative improvement of each method to the performance gap between the Pre-trained model and the Oracle model. From the results, we can infer that, 1) GDFM performs significantly better than compared methods. 2) It is noteworthy that FNW and ES-DFM brings a significant negative impact to NLL on the Taobao dataset, which is caused by the fake negative samples introduced by them; 3) The performance of FNW and MM-DFM has a more considerable variance compared with ES-DFM and GDFM (especially NLL on Taobao), since ES-DFM uses a fixed weighting model to re-weight its loss, which has a similar effect to GDFM's explicit regularizer that can stabilize training; 4) On the Criteo dataset, GDFM outperforms other methods without using post-click information other than conversion labels, which indicates that our information measure is effective for early conversions; 5) On the Taobao dataset, GDFM also utilizes post-click information such as cart and favorite. The results indicate that introducing rich post-click information into GDFM can improve the performance of CVR prediction.

### 4.3   Experimental analysis of GDFM

To further investigate the source of performance gain of GDFM, we construct several experiments. First, we remove the effect of information weights and the regularizer loss by setting the coefficients to zeros correspondingly. The results reported in Table. (2) indicate that the performance of GDFM is a combined effect of different components. Without $\alpha$ and $\lambda$, the performance drops significantly, which indicates that sample complexity and training variance are influential factors in the Criteo dataset. Dropping $\beta$ has relatively little influence to performance, which indicates that the temporal gap in the Criteo dataset has a weaker influence.

Secondly, to validate the capacity of GDFM to deal with more actions, we construct an additional user action in the Criteo dataset. This action is set to the same as $\mathbf{y}$ with a probability $p$, and set to $1 - \mathbf{y}$ with probability $1 - p$. In this way, we can investigate how the information measure influences the performance of GDFM. We report the results with different $p$ in Table. (3).

| Table 2: Effect of hyper-parameters on Criteo. | | | | Table 3: Adding more actions. | | | | |
|---|---|---|---|---|---|---|---|---|
| Method | AUC | PR-AUC | NLL | p | Entropy | AUC | PR-AUC | NLL |
| w/o $\alpha$ | 0.8333 | 0.6244 | 0.3983 | 0.500 | 0.529 | 0.835 | 0.631 | 0.396 |
| w/o $\beta$ | 0.8343 | **0.6314** | 0.3964 | 0.800 | 0.394 | 0.836 | 0.634 | 0.394 |
| w/o $\lambda$ | 0.8339 | 0.6226 | 0.3988 | 0.900 | 0.264 | 0.839 | 0.638 | 0.392 |
| GDFM(full) | **0.8349** | 0.6311 | **0.3960** | 0.950 | 0.166 | 0.840 | 0.641 | 0.391 |

When $p = 0.5$, the newly inserted action does not contain information about $y$, and the performance does not decrease, which indicates that GDFM can extend to more actions safely; When $p > 0.5$, the action becomes more informative, and the performance of GDFM steadily improves, which indicates that GDFM can utilize the information carried by actions efficiently; When $p = 0.95$, the action is highly informative to $y$, the performance approaches Oracle, which indicates that GDFM can efficiently extract information from high-quality action features.

Thirdly, we plot the weigths $w_{\text{info}}$, $w_{\text{time}}$ and $w$ corresponding to different revealing time $\delta$, with $\alpha = 2, \beta = 1$ as in other experiments. We can infer from the Figure. 4 that 1) the information carried by actions increases as the revealing time increases; 2) the temporal gap enlarges as the revealing time increases; 3) the combined information weights $w$ reaches its maximum value on the 7th day, which indicates that the information revealed on the 7th day is most informative to conversion prediction. Information revealed at other times still has non-negligible weights that influence training.

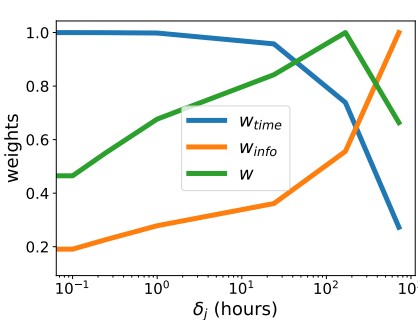

Figure 4: Information weights on the Criteo dataset.

## 5   Related work and discussion

Delayed feedback model (DFM) [5] firstly introduces survival analysis[25, 26] to deal with the delayed feedback problem. The following methods predict the probability of whether a user converts before a predefined waiting time[8–10]. However, how to define the strategy of duplicating training samples varies among previous literatures[8–10, 16]. On the contrary, the revealing time distribution is explicitly defined in GDFM, which indicates that each sample should be duplicated once at a revealing time. The information weight $\mathbf{w}$ in GDFM is effectively adjusting the revealing time with the aid of GDFM's information measure.

To the best of our knowledge, we are the first to analyze the property of learning with delayed feedback with a time-varying distribution. Existing analysis[5, 8, 9, 16] assumes the distribution is stable (unlimited samples from $p(\mathbf{x}, \mathbf{y})$)[5], or the action distribution $p(\mathbf{a}|\mathbf{x})$ can be estimated accurately[8, 9, 16]. Since $p^t(\mathbf{x}, \mathbf{y})$ indeed varies along with time (otherwise streaming training is unnecessary), assuming a static $p(\mathbf{x}, \mathbf{y})$ can not capture the property of the delayed feedback problem; The action distribution $p(\mathbf{a}|\mathbf{x}) = \sum_{\mathbf{y}} p(\mathbf{a}|\mathbf{x}, \mathbf{y}) p(\mathbf{y}|\mathbf{x})$ also changes following $p^t(\mathbf{y}|\mathbf{x})$. Thus, GDFM can be a better model for understanding and analyzing the delayed feedback problem.

Delayed feedback problem has also attracted attention from bandit learning communities[27–29], where the objective is to minimize regret in a decision problem. Vernade et al. [27] introduces a similar idea of using intermediate observations before getting rewards and shows that such intermediate observations can improve regret, which agrees with our analysis from a different perspective.

A limitation of GDFM is that the training cost grows linearly with the number of different revealing time $\delta_j$, which may be a potential bottleneck in large-scale streaming training.

# 6 Conclusion

We present an analysis of the delayed feedback problem based on the assumption that the relationships between post-click behaviors and conversions are relatively stable. Our results indicate that to improve the performance of learning under delayed feedback, we should utilize post-click information as complements. Therefore, we propose a generalized delayed feedback model to incorporate general user behaviors and a re-weighting method to utilize behavior information efficiently. Experiments on public datasets validate the effectiveness of our method.

## Acknowledgements

This work is supported by NSFC (61921006).

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
