# Generalized Delayed Feedback Model with Post-Click Information in Recommender Systems Supplementary Material

**Jia-Qi Yang**    **De-Chuan Zhan**[*]
State Key Laboratory for Novel Software Technology
Nanjing University, Nanjing, 210023, China
yangjq@lamda.nju.edu.cn, zhandc@nju.edu.cn

## 1  Proof of Lemma 3.1

Without loss of generality, we consider an action $\mathbf{a} \in \{a_1, ..., a_k\}$, and $p(\mathbf{a}|\mathbf{x}) = \sum_{\mathbf{y}} p(\mathbf{y}|\mathbf{x}) p(\mathbf{a}|\mathbf{x}, \mathbf{y})$, $\mathbf{y} \in \{y_1, ..., y_n\}$. Assuming we can estimate $p(\mathbf{a}|\mathbf{x})$ accurately, we have following results:

**Lemma 3.1.** *Use a matrix $\mathbf{M_x} \in \mathbb{R}^{k \times n}$ to denote the conditional probability $p(\mathbf{a}|\mathbf{x}, \mathbf{y})$, where $(\mathbf{M_x})_{ji} = p(\mathbf{a} = a_j | \mathbf{x}, \mathbf{y} = i)$. We can recover $p(\mathbf{y}|\mathbf{x})$ from $p(\mathbf{a}|\mathbf{x})$ if and only if $\mathrm{rank}(\mathbf{M_x}) = n$.*

*Proof.* We can denote $p(\mathbf{a}|\mathbf{x})$ as a vector $\mathbf{a}_x \in \mathbb{R}^k$, and $p(\mathbf{y}|\mathbf{x})$ as $\mathbf{y}_x \in \mathbb{R}^n$. So the value of $\mathbf{y}_x$ is determined by the linear equation system $\mathbf{M_x y}_x = \mathbf{a}_x$. We have $\mathrm{rank}([\mathbf{M_x}|\mathbf{a}]) = \mathrm{rank}([\mathbf{M_x}|\mathbf{M_x y}_x]) = \mathrm{rank}(\mathbf{M_x})$. By the Rouché-Capelli theorem[1], the solution of a system of the linear equations $\mathbf{M_x y}_x = \mathbf{a}_x$ is unique if and only if $\mathrm{rank}([\mathbf{M_x}|\mathbf{a}_x]) = \mathrm{rank}(\mathbf{M_x}) = n$. □

**Remark.** *In Lemma. 3.1, when $k < n$, we have $\mathrm{rank}(\mathbf{M_x}) \le k < n$, so the system of linear equations is undeterermined and there are infinitly many solutions.*

## 2  Conditional entropy and $D_{TV}$

We randomly generate distributions $p(\mathbf{y})$ and conditional distributions $p(\mathbf{a}|\mathbf{y})$ to investigate the relationship between conditional entropy and total variance distance. Specifically, we denote $\mathbf{y} \in [n]$, $\mathbf{a} \in [m]$, then we generate distributions in following steps:

1. We randomly generate a categorical distribution $p_1(\mathbf{y})$, then generate another distribution $p_2(\mathbf{y}) = p_1(\mathbf{y}) + \epsilon$, $p_2(\mathbf{y})$ is normalized. $\epsilon \sim \mathcal{N}(0, 0.05)$ is a small value that make sure the distance between $p_1(\mathbf{y})$ and $p_2(\mathbf{y})$ is close.

2. Check whether $0.05 - 0.005 \le D_{TV}(p_1(\mathbf{y}), p_2(\mathbf{y})) \le 0.05$, if not, drop this sample. In this way, we roughly sample $p_1(\mathbf{y})$ and $p_2(\mathbf{y})$ within distance range $[0.045, 0.05]$. Then we check the transformed distance range between $p_1(\mathbf{a})$ and $p_2(\mathbf{a})$.

3. We randomly generate conditional distributions $p(\mathbf{a}|\mathbf{y})$, and ensure that $\sum_{\mathbf{a}} p(\mathbf{a}|\mathbf{y}) = 1$, then we have $p_1(\mathbf{a}, \mathbf{y}) = p(\mathbf{a}|\mathbf{y}) p_1(\mathbf{y})$.

4. Calculate conditional entropy $H(\mathbf{y}_1|\mathbf{a})$. Note that since $p_1(\mathbf{y})$ and $p_2(\mathbf{y})$ are close, $H(\mathbf{y}_1|\mathbf{a})$ and $H(\mathbf{y}_2|\mathbf{a})$ have similar values, we use $H(\mathbf{y}_1|\mathbf{a})$ as $H(\mathbf{y}|\mathbf{a})$ in plots.

5. Calculate $p_1(\mathbf{a}) = \sum_y p(\mathbf{a}|\mathbf{y}) p_1(\mathbf{y})$, $p_2(\mathbf{a}) = \sum_y p(\mathbf{a}|\mathbf{y}) p_2(\mathbf{y})$.

6. Calculate $D_{TV}(p_1(\mathbf{a}), p_2(\mathbf{a}))$, which is reported in Figure. 1 as in paper. Figure. 1 reflects a rough relationship between $D_{TV}$ and $H$ with different $p(\mathbf{y})$ and $p(\mathbf{a}|\mathbf{y})$.

---

[*]De-Chuan Zhan is the corresponding author.

36th Conference on Neural Information Processing Systems (NeurIPS 2022).

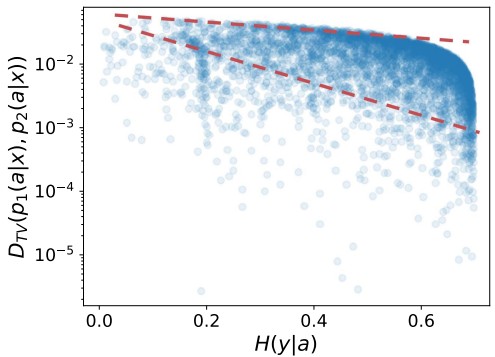

Figure 1: Conditional entropy and transformed distance.

In Figure. 1, we use $n = 2$ and $m = 2$, which corresponds to common case that conversions and actions are 01 valued. In general case, where $n > 2$ or $m > 2$, the total variation distance sill decreases along with increasing conditional entropy. However, we found that the relationship between conditional entropy and total variation distance is not strictly exponential, as depicted in Figure. 2 The relationship is worth further research.

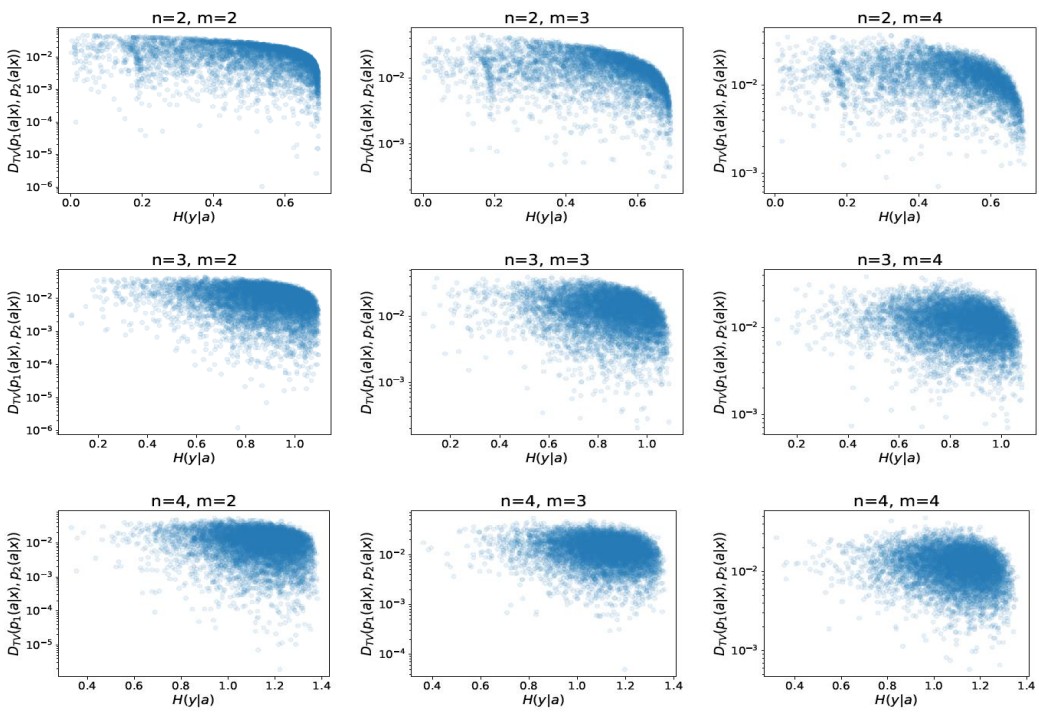

Figure 2: Conditional entropy and transformed distance with different $n$ and $m$

## 3 Reproduction

In this section, we describe the implementation details of GDFM and all the compared methods.

## 3.1 Dataset processing

**Criteo** There are 8 numerical features and 9 categorical features in the Criteo dataset. We use (64, 16, 128, 64, 128, 64, 512, 512) bins for numerical features, and (512, 128, 256, 256, 64, 256, 256, 16, 256) bins for categorical features. Each bin is represented with a 32-dimensional embedding. We found that increasing the number of bins or embedding size could not improve performance significantly. The revealing time distribution $p(\delta)$ is a uniform distribution on $\{0, 6min, 15min, 1hour, 1day, 7day, 30day\}$. We use data from the first 10 days to pre-train models, and data from the left 50 days to evaluate delayed feedback methods in streaming training.

**Taobao** We use 1000 bins for users, 10000 bins for items, and 1000 bins for item categories. The revealing time distribution $p(\delta)$ is a uniform distribution on $\{2min, 10min, 2hour, 1day, 3day\}$. We also add the last 5 user actions as a feature of the user, where each action consists of a list [(item, item category, action), ...]. If there are more than 5 actions, the earliest action is dropped; if the number of previous actions is less than 5, we pad with (0, 0, 0). We found that the original timestamp in the Taobao dataset is broken since most of the purchases happen before the corresponding click event, which must be inaccurate. Because the time interval of the Taobao dataset is too small (9 days), we think those purchases correspond to clicks happened before this time window. So we modify the timestamps to construct a realistic setting. Specifically, we adjust the purchase timestamps to be later than the first click of the user, and the delay time is enlarged. We reset the delay times that are longer than 3 days to 3 days. We use data from the first 2 days to pre-train models, and data from the left 7 days to evaluate delayed feedback methods.

## 3.2 Network and hyperparameters

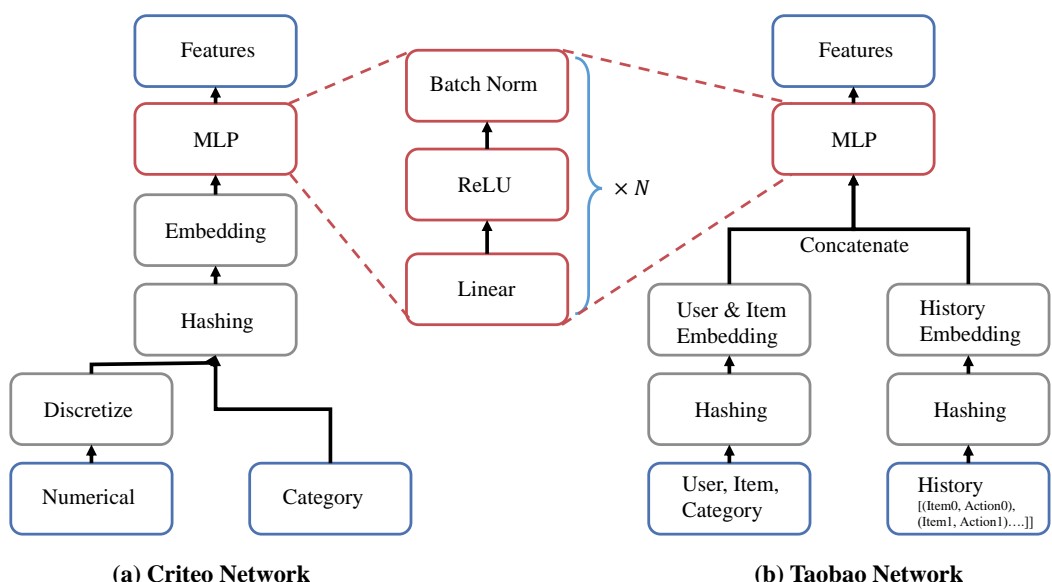

(a) **Criteo Network**         (b) **Taobao Network**

Figure 3: Feature networks on Criteo and Taobao datasets

The network architecture to compute features from $\mathbf{x}$ is depicted in Figure. 3. We use MLP with three layers using hidden sizes [256, 256, 128] on both Criteo and Taobao datasets. Following Weinberger et al. [2], we hash all the categorical features to reduce the number of different values. The CVR prediction model $p_\theta(\mathbf{x})$ is a feature network followed by a linear classification layer. The action distribution model $p_\phi(\mathbf{a}|\mathbf{x}, \mathbf{y}, \delta)$ use a feature network to extract feature of $\mathbf{x}$, then the feature of $\mathbf{x}$ is concatenated with one-hot representation of $\mathbf{y}$, the concatenated features are feed into a MLP corresponds to $p_\phi(\mathbf{a}|\mathbf{x}, \mathbf{y}, \delta)$. The output of $p_\phi(\mathbf{a}|\mathbf{x}, \mathbf{y}, \delta)$ has $m$ heads corresponds to $m$ different $\delta$, each head predicts a corresponding action $\mathbf{a}$.

**Hardware** We conduct all the experiments on a server with 370GB of memory, a Nvidia 3090Ti gpu, and two Intel(R) Xeon(R) Silver 4210R CPU. One pass of training and evaluation of each method can be conducted within 3 hours with our hardware.

**Hyperparameters** We use weight decay=1e-6, batch size=4096, learning rate=1e-3 in all the experiments; we use the Adam optimizer[3] and its default parameters in Pytorch[4]. The embedding size of each bin is 32 in both Crito and Taobao datasets.

### 3.2.1 Method specific implementations

**MMDFM** To make a fair comparison, we use the same revealing time as GDFM.

**ESDFM** We use elapsed time of 0.25 hours as suggested by the authors[5] on the Criteo dataset. We use elapsed time 1 hour on the Taobao dataset.

**GDFM** We use $\alpha = 2$, $\beta = 1$, $\lambda = 0.01$ in all the experiments. Since there are multiple different $\delta$, the actions that happen earlier can be used multiple times during training. Specifically, if $\delta_j < \delta_{j+1}, 1 \leq j < m$, actions revealed at $\delta_j$ will be observed $m - j + 1$ times. To utilize available information at different times, we use all the observed actions in training and introduce a weight $\frac{1}{m-j+1}$ on revealing time $\delta_j$. This weight is combined with the information weights $w$ during training. Since we use a multi-head network to predict $p(\mathbf{a}|\mathbf{x}, \mathbf{y}, \delta)$, reusing observed actions will not incur additional training overhead. This trick is also applied to MMDFM to ensure a fair comparison but is not applicable to importance sampling based methods such as FNW and ESDFM.

### 3.3 Raw results

In the paper, we report the relative performance of each method. The raw data can be recovered from relative performance. We provide the raw data in Table. (1) and Table. (2) for convenience.

Table 1: Raw data on Criteo dataset

| Method | Criteo | | | Criteo-STD | | |
|---|---|---|---|---|---|---|
| | AUC | PR-AUC | LL | AUC | PR-AUC | LL |
| Pretrain | 0.81508 | 0.60762 | 0.41417 | 0 | 0 | 0 |
| Vanilla | 0.82175 | 0.61643 | 0.40805 | 6.62E-05 | 7.64E-05 | 3.36E-05 |
| FNW | 0.83151 | 0.62274 | 0.40415 | 9.12E-05 | 0.000158415 | 0.000304291 |
| ESDFM | 0.834007 | 0.62981 | 0.3976914 | 0.00014051 | 0.00039714 | 0.0002975 |
| MMDFM | 0.833549 | 0.621357 | 0.400679 | 0.0003229 | 0.002348 | 0.000904 |
| **GDFM** | 0.83492 | 0.63151 | 0.39614 | 0.00017247 | 0.00057246 | 0.0001569 |
| Oracle | 0.84158 | 0.64268 | 0.38928 | 5.44E-05 | 0.000122832 | 5.69E-05 |

Table 2: Raw data on Taobao dataset

| Method | Taobao | | | Taobao-STD | | |
|---|---|---|---|---|---|---|
| | AUC | PR-AUC | LL | AUC | PR-AUC | LL |
| Pretrain | 0.703124 | 0.054323 | 0.084635 | | | |
| Vanilla | 0.715412667 | 0.059863333 | 0.083957667 | 0.000165458 | 0.000150557 | 2.80159E-05 |
| FNW | 0.7113876 | 0.0515844 | 0.0899612 | 0.00015735 | 0.000152984 | 0.0001897 |
| ESDFM | 0.7161926 | 0.056079 | 0.0877858 | 0.000145676 | 0.000287079 | 1.73628E-05 |
| MMDFM | 0.7156006 | 0.0599014 | 0.0848546 | 0.000757665 | 0.000231189 | 0.000154145 |
| **GDFM** | 0.719688 | 0.0615742 | 0.0839052 | 9.71785E-05 | 7.96856E-05 | 4.58745E-05 |
| Oracle | 0.723986 | 0.063303 | 0.083163 | | | |

# 4 Discussion on an alternative approach

We discuss an alternative definition of GDFM in this section. Intuitively, users may change their minds after clicks, which may be disclosed by the user behaviors. So the conversions should depend on user behaviors. This view suggests the following definition

$$p^t(\mathbf{x}, \mathbf{y}, \mathbf{a}, \delta) = p^t(\mathbf{a}|\mathbf{x}, \delta)p(\mathbf{y}|\mathbf{x}, \mathbf{a}, \delta)p(\delta)p^t(\mathbf{x}) \tag{1}$$

Recall our definition of GDFM in the paper

$$p^t(\mathbf{x}, \mathbf{y}, \mathbf{a}, \delta) = p^t(\mathbf{y}|\mathbf{x})p(\mathbf{a}|\mathbf{x}, \mathbf{y}, \delta)p(\delta)p^t(\mathbf{x}) \tag{2}$$

There are several differences between Eq. (2) and Eq. (1).

1. In Eq. (1), the conversion label $\mathbf{y}$ depends on $\mathbf{a}$. On the contrary, the conversion labels do not depend on $\mathbf{a}$ in Eq. (2). This difference reflects how we explain user behaviors: In Eq. (2), user behaviors are triggered by their conversion intentions, and in Eq. (1), the conversion intentions change along with user behaviors. Both formulations model a relationship between conversion labels and post-actions, and the relationship is utilized to improve the target model. So it is hard to say which one could model the delayed feedback problem better without considering the following perspectives.

2. To predict conversion rates, the alternative approach should sum on $\mathbf{a}$ and $\delta$ as follows

$$p^t(\mathbf{y}|\mathbf{x}) = \sum_{\mathbf{a}, \delta} p^t(\mathbf{a}|\mathbf{x}, \delta)p(\mathbf{y}|\mathbf{x}, \mathbf{a}, \delta)p(\delta) \tag{3}$$

   In Eq. (3), the $p^t(\mathbf{a}|\mathbf{x}, \delta)$ and $p(\mathbf{y}|\mathbf{x}, \mathbf{a}, \delta)$ should be estimated with neural networks. The summation is on possible actions $\mathbf{a}$ and revealing time $\delta$. Thus, the computational complexity scales as $O(k)$ ($k$ different actions), which is a fatal drawback of this approach. On the contrary, predicting with GDFM in Eq. (2) does not incur additional computational burden since the model of $p(\mathbf{y}|\mathbf{x})$ is directly available.

3. Streaming training of the alternative approach in Eq. (1) is cheaper than GDFM since $p(\mathbf{a}|\mathbf{x})$ is modeled with a neural network, so we do not need to sum over $\mathbf{y}$ in streaming training. However, since the number of possible values of $\mathbf{y}$ is typically small (01 valued in common cases) compared with actions ($k$), the computational overhead of GDFM is small.

4. Another drawback of the alternative definition in Eq. (1) is that the actions model $p(\mathbf{a}|\mathbf{x})$ can influence predictions directly. As our analysis in the paper, a non-informative action may be harmful to the conversion prediction task, and we propose a method to make GDFM safer. However, it is unclear how to alleviate the influence of action in the alternative approach since the action model and conversion model are entangled together.

# 5 Existing methods in GDFM framework

**DFM[6]** In the delayed feedback model (DFM), elapsed time $e$ corresponds to the revealing time $\delta$ in GDFM. DFM denotes the observed conversion label as $o$ and the ground-truth conversion label as $y$. DFM also defines a delay time $d$, which is the delay of conversion after click. DFM does not define the data distribution explicitly but defines several conditional distributions instead. The probability that a sample $x$ is observed as positive at elapsed time $e$ is defined by

$$p(o = 1|x, e) = p(o = 1|x, e, y = 1) = p(e \geq d|x, y = 1)p(y = 1|x) \tag{4}$$

which utilizes the fact that an observed positive must be a positive sample. The probability that a sample $x$ is observed as negative at elapsed time $e$ is defined by

$$p(o = 0|x, e) = p(y = 0|x) + p(y = 1|x)p(e < d|x, y = 1) \tag{5}$$

Where $p(d|x, y = 1) = \lambda(x) \exp(-\lambda(x)d)$, so $p(e < d|x, y = 1)$ can be calculated analitically.

In GDFM, the observed label $o$ corresponds to action $a$. Thus, $p(o|x, e)$ corresponds to $p^t(a|x, \delta) = \sum_y p(a|x, y, \delta)p^t(y|x)$. DFM follows a case by case deduction assuming that actions and conversions are 01 valued. On the contrary, GDFM is more clear and supports multi-class actions and conversions.

The representations of other DFM-based methods such as MM-DFM[7] and KDE-DFM[8] in the GDFM framework are similar.

**ES-DFM[5]**   In ES-DFM, the observed label distribution is corrected by an importance sampling method[9]

$$\mathcal{L}_{ideal} = \mathbb{E}_{(x,y)\sim p(x,y)}\ell(y, f_\theta(x)) \tag{6}$$

$$= \int p(x)dx \int p(y|x)\ell(y, f_\theta(x))dy \tag{7}$$

$$= \int p(x)dx \int q(y|x)\frac{p(y|x)}{q(y|x)}\ell(y, f_\theta(x))dy \tag{8}$$

$$\approx \mathbb{E}_{(x,y)\sim q(x,y)}\frac{p(y|x)}{q(y|x)}\ell(y, f_\theta(x)) \tag{9}$$

$$= \mathcal{L}_{iw} \tag{10}$$

where $q(y|x)$ models the observed label distribution $p(o|x)$, which is estimated with a neural network. $q(y|x)$ corresponds to the action distribution $p^t(a|x)$ in GDFM. In this view, since $p^t(a|x) = \sum_y p(a|x, y)p^t(y|x)$, where $p^t(y|x)$ changes along with time, the estimation of $q(y|x)$ can hardly be accurate (since the latest data is from $p^{t-\delta}(a|x)$).

The representations of other importance sampling based methods such as FNW[10] and DEFER[11] in the GDFM framework are similar.

**Advantages of GDFM**   From the above analysis, we can see that existing methods lack the consideration of the distribution change along with time $t$, which is important in the delayed feedback problem. By introducing time-dependent data distribution $p^t$, GDFM enables a more realistic analysis of the delayed feedback problems in streaming training (section 3.2 in paper), which may also be utilized to improve importance sampling based methods.

## 6   Computational complexity

Algorithm 2 is to calculate the joint distribution $p(a, y)$, which can be achieved by an $O(N)$ ($N$ is the number of samples) counting over the dataset. Since we assume the distribution $p(a, y)$ is relatively stable, we only need to run the algorithm once on an offline dataset. Thus, the computational complexity of Algorithm 2 is negligible and will not affect the streaming training stage.

The main increase of computational complexity is caused by Algorithm 1.

1. Introducing multiple revealing times requires to insert multiple duplicated samples into the data stream. This leads to O(number of different revealing times) increase of training data, which means the number of FLOPS will also increase by the same scale. Since the primary problem is the lack of timely label, and the increase of data can be greatly alleviated by data parallel, the overall cost is affordable.

2. The calculation of the GDFM loss (Eq. 10 in paper) can be achieved by one pass of the feature network, so the computational complexity of calculating the GDFM loss is roughly the same as plain CVR loss (cross entropy). Specifically, Calculating $q(a|x, y, \delta)$ needs one forward pass of the feature network, which can be shared with $q(y|x)$ as depicted in Figure 1 of the revised paper. We use multiple output heads to produce predictions of different $\delta$, so we can get all the necessary predictions with two forward passes ($q(a|x, y = 0, \delta)$ and $q(a|x, y = 1, \delta)$). We can also deal different $y$ in the same way as $\delta$ by adding more output heads, which can reduce the time complexity of calculating $q(a|x, \delta) = \sum_y q(a|x, y, \delta)q(y|x)$ to the same scale of the original CVR loss.

Thus, the overall computational burden is O(number of revealing times) of duplicated data, which can be directly parallelized.