# OpenReview forum: "Generalized Delayed Feedback Model with Post-Click Information in Recommender Systems"
_NeurIPS.cc/2022/Conference — NeurIPS 2022 Accept_

### Official Review · Reviewer_fM2F · 2022-06-29

**Rating:** 7
**Confidence:** 4
**Soundness:** 3 good
**Presentation:** 3 good
**Contribution:** 3 good

**Summary:**

This paper aims to alleviate the delayed feedback problem in conversion rate prediction (CVR) via proposing a new method: generalized delayed feedback model (GDFM). After clicking the item url, it might take months for a user to really purchase the item. Hence, the positive sample might be treated as negative sample in the training and this problem could bias the model.

Instead of only relying on observed labels, their key idea is to introduce other user behaviors or actions like put the item into shopping cart as an auxiliary information or proxy to better predict the conversion.

According to entropy between the conversion label and an action, they design a weight to reflect how informative the action is. Besides, they also consider the time gap and sample complexity into designing the weight for the action.

They conduct experiments over two public datasets and propose detailed evaluations.

**Questions:**

(1) Using entropy between user action and conversion label to measure the information carried by the action seems reasonable. In addition, they also consider time gap and sample complexity into designing the weight for the action, which shows the comprehensiveness of their thinking. My question (also presented in the next section) is that are these factors firstly considered by this paper? did any related work consider these factors?

(2) In 167: "Since rank(M) < n requires that some rows of M are linear combination of the others,  which rarely happens in real-world problems". So you are saying rank(M) = n holds for most of real-world problems, may I ask why?

In this specific situation, what is the physical meaning of rank(M) = n? does that mean user action and conversion has a very strong relationship? like a user must purchase the item in the shopping cart?

in 160, "We can recover p(y|x) from p(a|x) if and only if rank(Mx) = n." My opinion is that rank(Mx) = n is a very strong condition and also we don't need to 100% recover p(y|x) from p(a|x), does I understand right?

(3) In section 3.2.2, I understand that how much information an action carry about the conversion can be measured by the entropy between them and I see H (y|a) in Equation (5). However, in 195: "Since we are estimating p(y|x) through a proxy distribution p(a|y)". May I ask where does p(a|y) come from? could you show the relation of p(y|x), p(a|y) and p(a|x) in some sort of equations, I am a bit confusing here.



**Limitations:**

In Section 2, they said: "However, the ground-truth conversion labels are available only after a long delay: A user may purchase several months later after a click event x."

Well, several months? that is a bit against my life experience.

Does this delay of several months a common situation?

I guess it would be better to show the delay distribution from a real-world CVR system so that we know how serious this delay problem can be and how important this paper's contribution can be.

But I understand that it may not be appropriate to show that due to company's policy.

Expected to see that though!

**Strengths And Weaknesses:**

Strengths:

(1) Since years ago, papers of recommender systems have become some sort of LEGO game -- like how to put a CNN over a DNN etc., Unlike those papers, this one really analyze the impact of user actions and clearly present the generation process in a probability graph way, which looks reasonable to me. I really enjoy this kind of papers.

(2) Using entropy between user action and conversion label to measure the information carried by the action seems reasonable. In addition, they also consider time gap and sample complexity into designing the weight for the action, which shows the comprehensiveness of their thinking. My question (also presented in the next section) is that are these factors firstly considered by this paper?

Weakness:

(1) The experiments could be further enhanced:

(a) Is there any online experiments?

(b) What is the architecture used in the experiments? there are no clues in the paper.

(2) Will algorithm 2 significantly slower the training? Where is the time complexity analysis?

---

> ### Author Response · Authors · 2022-08-02
> **Response to Reviewer fM2F**
>
> Thanks for your kind and patient review of our paper.
> We try our best to answer each question in detail within the character limit.
>
> ### Q1: Are there any online experiments?
>
> We have conducted offline experiments on private datasets from our cooperative partner,
> and the results are promising to prompt us to an online deployment.
> However, the development is not yet complete, and we are still working on it.
>
> ### Q2: What is the architecture used in the experiments?
>
> We use hashed user ID and item ID to train embeddings end-to-end,
> then the embeddings are concatenated to form the input of a MLP,
> the outputs of this MLP serve as embeddings of $x$.
> We have included a new figure of the overall framework in the revision of our paper,
> and the feature encoder network is depicted in the supplementary material.
>
> ### Q3: Time complexity analysis?
>
> Algorithm 2 is to calculate the joint distribution $p(a, y)$,
> which can be achieved by an $O(N)$ ($N$ is the number of samples) counting over the dataset.
> Since we assume the distribution $p(a, y)$ is relatively stable,
> we only need to run the algorithm once on an offline dataset.
> Thus, the computational complexity of Algorithm 2 is
> negligible and will not affect the streaming training stage.
>
> The main increase in computational complexity is
> caused by Algorithm 1: Introducing multiple revealing times requires to
> insert multiple duplicated samples into the data stream.
> This leads to an O(number of different revealing times) increase
> of training data.
>
> Thus, the overall computational burden is O(number of revealing times)
> of duplicated data.
> Since the primary problem is the lack of timely labels,
> and the increase of data can be greatly alleviated
> by data parallel, the overall cost is affordable.
>
> ### Q4: Are there any related work considered ideas similar to the paper?
>
> * Using entropy between user action and conversion label to measure the
>   information carried by the action.
>
> To the best of our knowledge, we are the first to use entropy
> to measure the information carried by the actions.
>
> * Considering the time gap and sample complexity into designing the weight for the action.
>
> Such a problem only matters in online learning with delayed feedback,
> and existing literature does not consider the distribution drift along with time.
> Some related work also involves losses corresponding to different actions,
> but they use equal weights[2] or treat the weights as independent hyper-parameters[1].
>
> [1] Chen, et al. Efficient Heterogeneous Collaborative Filtering without Negative Sampling for Recommendation. AAAI'20.
>
> [2] Ma, et al. "Entire space multi-task model: An effective approach for estimating post-click conversion rate." SIGIR'18.
>
> ### Q5: Why does the assumption "$rank(M) = n$" holds for real-world problems?
>
> Considering conversion $y$ and cart action $a$.
> Using the fact that $p(a=0|y)+p(a=1|y)=1$,
> we can solve that $p(a=0|y=0)=p(a=0|y=1)$ and $p(a=1|y=0)=p(a=1|y=1)$ must hold **exactly** if $rank(M) < n$,
> which is nearly impossible naturally.
>
> $rank(M)=n$ is a necessary but not a sufficient condition to make $p(y|x)$ recoverable (with low error),
> and it does not imply a strong relationship.
> The reason is that the estimation of $p(a|x)$ is not perfect,
> and the error will be amplified by $M$.
> So we further propose to use conditional entropy as a measure of relationship strength.
>
> ### Q6: The relationship between $p(y|x)$, $p(a|y)$, and $p(a|x)$.
>
> By definition, for a fixed $\delta$ (omitted in the following equations),
> we have
> $$
> p(a|x) = \sum_y p(a|y, x) p(y|x)
> $$
> and practically we approximate $p(a|y, x)$ with a $q(a|y)$ which
> does not depend on $x$. We will further clarify this point in the paper.
> Here, we are maximizing the likelihood of $p(a|x)$ to
> learn $p(y|x)$ *using $q(a|y)$ as a bridge*,
> so $q(a|y)$ works like a proxy between $p(a|x)$ and $p(y|x)$.
>
> ### Q7: The significance of the delayed feedback problem in real-world recommender systems.
>
> Extremely long delay (several months) does exist in real-world recommender systems (but rare).
> Practically, delay from hours to 1 week occupies a considerable portion,
> and we provide some statistics from our partner and some previous literature here:
>
> 1. E-commerce (our partner): With $\delta_y=14$ days,
>    28% of the conversions happen after 1 hour, and 15% of conversions happen after 1 day.
>    Our preliminary online experiments indicate that simply dropping the conversions with > 1-hour delay
>    leads to significant performance improvement,
>    which is indeed a degenerated case of GDFM with only one revealing time.
>
> 2. Display advertisement (Criteo[1]): With $\delta_y=30$ days, 70% of conversions happen after 1 hour,
>    50% of conversions happen after 1 day, and 13% of conversions happen after 14 days.
>    [1] provides more detailed information.
>
> [1] Chapelle, et al. Modeling Delayed Feedback in Display Advertising. KDD'14.
>
> The Reviewer fuKP also agrees that "This is a very important problem."

---

> > ### Comment · Reviewer_fM2F · 2022-08-05
> > **More question**
> >
> > About Q6, q(a|y)? or p(a|y)?
> >
> > OK, practically you approximate p(a|y, x) with a p(a|y), how much information will be lost in such approximation may I ask?

---

> > > ### Author Response · Authors · 2022-08-06
> > > **Response to More question**
> > >
> > > We use $p()$ to denote the ground-truth probability distribution,
> > > and $q()$ to denote the corresponding estimated probability distribution.
> > > Here, $q(a|y)$ is an estimation of $p(a|y, x)$.
> > > Sorry for the confusion.
> > >
> > > It's hard to tell whether using $q(a|y, x)$ will be better.
> > > Since the data with specific $x$ and $y$ will be much fewer,
> > > which may lead to overfitting and larger approximation error.
> > >
> > > Both $q(a|y)$ and $q(a|y, x)$ formulation fit within our framework,
> > > and the choice depends on the specific problem.
> > > Experimentally, we found that using $q(a|y)$ is simpler and performs well in our case.

---

> > > > ### Comment · Reviewer_fM2F · 2022-08-08
> > > > **feedback**
> > > >
> > > > Thanks for the clarification.
> > > >
> > > > I have raised the score to 7.

---

### Official Review · Reviewer_RiVF · 2022-07-12

**Rating:** 3
**Confidence:** 5
**Soundness:** 2 fair
**Presentation:** 2 fair
**Contribution:** 2 fair

**Summary:**

In this paper, based on the assumption that post-click user behaviors are informative to conversion rate prediction and can be used to improve timeliness, the authors propose a generalized delayed feedback model (GDFM) that unifies both post-click behaviors and early conversions as stochastic post-click information. Based on GDFM, they further establish a novel perspective that the performance gap introduced by delayed feedback can be attributed to temporal and sampling gaps. They measure the quality of post-click information with a combination of temporal distance and sample complexity. The training objective is re-weighted accordingly to highlight informative and timely signals. The experimental performance results show their model’s effectiveness.

**Questions:**

Please answer my questions corresponding to the negative points: (1) Model novelty; (2) Missing baselines.
See the detailed comments above.

**Limitations:**

Some limitations are mentioned and I think such this algorithm-driven paper for improving user experience in recommendation system does not have negative societal impact.

**Strengths And Weaknesses:**

*Positive points*
1. Recommendation is one of the most important applications of machine learning.
2. The proposed method achieves good performance.
3. The method is proposed based on a thorough analysis.

*Negative points*
1. The proposed method is not novel enough. The authors assert that previous literature concentrates on utilizing early conversions to mitigate delayed feedback problem and propose a generalized delayed feedback model (GDFM) that unifies both post-click behaviors and early conversions as stochastic post-click information. Indeed, what they tend to address can be defined as heterogeneous feedback problem and there are some relationships between different feedback types. Besides, there have already been some works such as EHCF that addressed this problem via multi-task learning and linear transformation, whose solution is very similar to this work.

[1] Chen et al. Efficient Heterogeneous Collaborative Filtering without Negative Sampling for Recommendation. AAAI 20.

2. The baselines are not enough. It is necessary to compare with some important and highly related heterogeneous collaborative filtering recommendation models such as EHCF.

3. The presentation should also be improved. For example, there is not model framework in their methodology part as their proposed method is with multi-target. I strongly suggest a figure that can illustrate their overall design.

---

> ### Author Response · Authors · 2022-08-02
> **Response to Reviewer RiVF**
>
> Thanks for your detailed review and suggestions on improving the paper.
> We try our best to answer each question in detail within the character limit.
>
> ### Q1: What are the differences between the delayed feedback problem and the heterogeneous feedback problem?
>
> 1. **The training schema is different**.
>    Streaming training with feedback delay already differs
>    from static training without considering heterogeneous labels.
>    For example, in offline training, the missing labels will not be revealed during training,
>    and we do not need to consider conflict labels (e.g., negative label changes to positive label).
>
> 2. **Some distinct problems arise from feedback delay**.
>    Tackling new labels is not as straightforward as it seems to be at the first glance.
>    1. Suppose a sample has been already used as a negative sample, and it converts later,
>       *how* to deal with this sample?
>       If we simply ignore it, then we have used the wrong label;
>       if we insert a duplicate with a positive label, then the data distribution $p(x)$
>       changes (negative samples appear once, but positive samples may appear twice),
>       and the label conflict still exists, how to repair it?
>    2. Another problem is *when* to use a sample.
>       Since we are not working with a static dataset passively,
>       we can choose the revealing time freely,
>       and this requires us to define a schedule explicitly for revealing the labels.
>
> 3. **User actions play an intrinsically different role in learning with delayed feedback**:
>    In the setting of learning with heterogeneous feedback,
>    as we discussed in response to Reviewer fM2F Q4, user actions work more like
>    *complementary information to the conversion labels*.
>    In the setting of learning with delayed feedback,
>    we need to *rely on user actions to extract information related to conversions*
>    when ground-truth conversion labels have not been revealed yet.
>
> These distinct differences lead to different motivations and methodologies of
> delayed feedback methods compared with heterogeneous feedback methods.
>
> ### Q2: What are the differences between GDFM and EHCF?
>
> 1. EHCF does not consider specific problems in learning with delayed feedback as
>    discussed in the previous question.
>
> 2. As pointed out by Reviewer fM2F and fuKP, our main contribution is providing a
>    novel probabilistic perspective to analyze the delayed feedback problem,
>    and come up with a practical method to measure the information carried by
>    user actions. These are novel points rooted in the delayed feedback problem and
>    are not considered by EHCF.
>
> 3. We agree that in our current implementation of GDFM,
>    $p(a|x)$ also relates with $p(y|x)$ linearly.
>    However, this formulation comes from our probabilistic model naturally
>    with clear interpretability and establishes the base of the following analysis.
>    The linear mapping introduced by EHCF lacks such probabilistic insight.
>
> 4. The training methods are different, which leads to different results:
>    GDFM learns $p(a|y)$ explicitly,
>    whereas the meaning of linear transformation learned by EHCF
>    is unclear.
>
> ### Q3: Experimental comparison between GDFM and heterogeneous feedback methods such as EHCF.
>
> As analyzed in the previous questions,
> we can not compare GDFM with EHCF directly since EHCF does not support duplicated samples and
> changing labels in the delayed feedback setting.
>
> We agree that using trainable linear layers to capture relationships between user actions
> is an applicable idea in the delayed feedback setting with user actions.
> So we implemented an architecture equipped with the Transfer-based Multi-Behavior Prediction layer
> as proposed in the EHCF paper
> and used the same duplicating and revealing strategy as in GDFM to conduct a reasonable comparison.
> We denote this as the "Linear relation" method.
> We evaluate the performance on the Taobao dataset.
>
> The performance of the Linear relation method is:
>
> AUC: 63.4±0.9%, PR-AUC: 50.1±1.5%, NLL: -470±4.6%
>
> and GDFM is:
>
> AUC: 79.4±0.5%, PR-AUC: 80.7±0.9%, NLL: 49.6±3.1%
>
> The results support that utilizing the relationship between user actions and conversions with a proper sampling strategy
> will improve performance on AUC and PR-AUC.
> However, since the Linear relation method does not consider label changing,
> the NLL is significantly worse. And the AUC and PR-AUC metrics of GDFM are also better than EHCF based method.
>
> The experimental results and discussion about some related heterogeneous feedback methods [1, 2, 3]
> are added to the paper.
>
> [1] Chen, et al. Efficient Heterogeneous Collaborative Filtering without Negative Sampling for Recommendation. AAAI'20.
>
> [2] Chen, et al. Graph Heterogeneous Multi-Relational Recommendation. AAAI'21.
>
> [3] Jin, et al. Multi-behavior Recommendation with Graph Convolutional Networks. SIGIR'20.
>
> We have added a new figure depicting our approach's overall design and procedure in
> revision of our paper according to your suggestion.

---

> ### Comment · Reviewer_fM2F · 2022-08-05
> **Heterogeneous feedback vs. the delayed feedback**
>
> Hi, I checked this paper "Heterogeneous Collaborative Filtering without Negative Sampling for Recommendation. AAAI 20."
> I don't think Heterogeneous feedback and the delayed feedback (this submission) are the same problem.
> Could you provide more thoughts?

---

### Official Review · Reviewer_fuKP · 2022-07-17

**Rating:** 6
**Confidence:** 3
**Soundness:** 3 good
**Presentation:** 3 good
**Contribution:** 2 fair

**Summary:**

The authors introduce a generalized feedback model in order to use delayed labels. This is very relevant in e-commerce platforms where a specific action we care about (for example buying a product) can happen after some time from the first interactions with the platform. Generally this is an important problem in e-commerce platforms. Is related basically, to attribution mechanism which is very important when one considers to build a training dataset from click-stream data. The authors compare their approach with other similar approaches on to two datasets. The authors observe improvements using their approach.

**Questions:**

- I am not sure I got on how many actions you consider in your experiments. In on-line platforms we can have quite large number of different actions. How you deal with this?
- Did not get in the architecture when you say that you concatenate one-hot encoded y. For y \in {0, 1} what exactly you concatenate? Could you clarify?
- I see in the experiments you use a quite large batch size. How you came up with this value?

**Ethics Review Area:**

["I don’t know"]

**Limitations:**

Yes the authors discuss briefly limitations and giving some idea of overcoming it, but they do not really elaborate on the type of the solution.

**Strengths And Weaknesses:**

This is very important problem. The way is tackled by the authors is interesting and they try to clarify all the points of their approach. By adding and modeling the intermediate actions they can capture and propagate the delayed feedback.

There is novelty in the approach as it considers temporal distributions and is the main one.

I have some doubts of whether such an approach can be easily implemented in a real use case. No such on-line experiments are described in the paper and essentially the paper builds on such a use-case. By also checking the exact results in the supplemental material improvements there is some relatively small improvements.

---

> ### Author Response · Authors · 2022-08-02
> **Response to Reviewer fuKP**
>
> Thanks for your kind and patient review of our paper.
> We try our best to answer each question in detail within the character limit.
>
> ### Q1: How many actions do you consider in your experiments?
>
> In the Criteo dataset, we consider one type of user action, i.e., conversion,
> with 7 different revealing times as described in the supplementary material,
> so there are 1*7=7 actions (we count different revealing times as different actions to simplify discussion).
>
> In the Taobao dataset, we consider 3 types of actions, i.e., conversion, cart, and favorite,
> with 5 different revealing times, so there are 3*5=15 actions.
>
> ### Q2: How will you deal with a large number of different actions?
>
> The computational complexity increase is mainly caused by the number of different revealing times,
> as analyzed in the response to Reviewer fM2F Q3.
>
> The reason is that practically we can utilize all the available action information at
> the same revealing time, which only needs forward and backward propagation once.
> And the computational increase caused by the number of outputs (actions) is
> typically negligible compared with the feature network.
>
> ### Q3: How do you deal with discrete input values (y)?
>
> We have added a new Figure 1 to depict the calculating procedure visually.
>
> The procedure for calculating equation (4) is:
>
> 1. The CVR probability $q_{\theta}(y|x)$ is calculated normally with one forward pass of network $q_{\theta}$,
>    which produces the estimation of $p(y=0|x)$ and $p(y=1|x)$.
>
> 2. $q_{\phi}(a_j|x, y, \delta_j)$ takes $x$, $\delta_j$ and $y$ as inputs.
>    Specifically, we first encode $x$ with an encoding network $Encode(x)=e_x$,
>    where $e_x$ denotes an embedding of $x$.
>    Then, we concatenate $e_x$ with one-hot representations of $y$ ([1, 0] and [0, 1] for CVR),
>    respectively, e.g., $e_x | [1, 0]$ and $e_x | [0, 1]$ (since we need to take sum over different $y$).
>    We take $e_x | [0, 1]$ (corresponds to $y=1$) as an example,
>    this vector is then fed into a MLP with $m$ output heads that corresponding to probabilities of $m$ actions.
>
> 3. The predicted probabilities are used to calculate the GDFM loss $\mathcal{L}_{\delta_j}$.
>
> ### Q4: Why do you use a large batch size?
>
> We tune the batch size on the vanilla method and keep it fixed when we compare other methods.
> Our practice suggests that a very small batch size will lead to severe overfitting to very recent data
> and will slow down training speed significantly.
>
> ### Q5: Elaborate on the limitations and potential solutions.
>
> As stated in the paper,
> the main limitation of our approach is the increase of data samples,
> as analyzed in the response to Reviewer fM2F Q3.
> Since the main problem is the lack of conversion labels,
> such an increase in computational complexity is acceptable.
> Besides, our current implementation utilizes data parallel training,
> which will not be significantly slower as long as the computational resource is sufficient.
>
> We can also subsample the data stream randomly to reduce computational burden if necessary,
> which is a common practice in recommendation systems.
>
> ### Q6: Consideration of the implementation complexity of the method.
>
> The training procedure of GDFM can be implemented efficiently with
> vectorized computation as in our code.
> There may be some engineering problems related to constructing the required data stream.
> Specifically, our approach needs to duplicate a click sample at each revealing time
> and get corresponding action labels,
> which is technically tractable but not quite compatible with some existing offline training frameworks.
>
> ### Q7: The experimental improvements are relatively small.
>
> The absolute improvements of GDFM compared with the best baselines
> are about 0.1% AUC, 0.17% PR-AUC on the Criteo dataset,
> and 0.35% AUC, 0.2% PR-AUC on the Taobao dataset,
> which is generally considered significant in recommender systems.
> And the improvements are statistically significant with $p \ll 0.01$.
>
> With the development of machine learning based recommender systems in recent years,
> the potential room for improvement restricted within the static dataset is significantly reduced,
> which is the main force driving the emergence of research considering delayed feedback.
> In learning with delayed feedback, the room for improvement lies between the static model and the
> Oracle model (streaming training without delayed feedback),
> which is about 2% AUC in our cases.
> The absolute improvement of GDFM will improve as this delayed feedback gap enlarges.
> For example, we observe that the delayed feedback problem is
> more severe on the items with a high price,
> which requires more time to decide to pay;
> the average feedback delay will also increase if we use larger $\delta_y$.

---

### Meta-Review · Area_Chair_uL1i · 2022-08-29

**Recommendation:** Accept
**Confidence:** Less certain

**Metareview:**

The paper presents an approach for dealing with delayed feedback in online learning settings such as large scale recommender systems, where the delays may be significant as in the case of predicting conversion rate for online shopping where a user may spend days or weeks deciding to finally click "purchase" after first viewing listings or putting items in a shopping cart.

There were quite a range of viewpoints on this paper, including a clear rejection from RiVF.  After reading through the paper myself and the responses from the authors and other reviewers to the points raised by this reviewer, I have determined that the key argument made by this reviewer -- that the paper is not novel, due to prior work on Efficient Heterogeneous Collaborative Filtering -- is not sufficient to warrant rejection of the paper.  The GDFM and EHCF settings are fundamentally different, and while one could imagine using methods from EHCF as part of a solution in this problem space, the current paper focuses clearly on the inherent difficulty of delayed feedback.  Furthermore, as the other two reviewers note, the problem itself is highly important, difficult to solve, and the current paper puts forward interesting and effective methods that are reasonably evaluated on publicly available static datasets.  For these reasons, I am discounting the rejection and recommending acceptance.

I do agree with reviewers who suggest that online ("real world") experiments would strengthen the paper significantly.  I understand that the nature of production level / deployed industrial settings can make it difficult to exhaustively report results, but even a paragraph of anecdotal evidence or experience here would be helpful.  I believe that the paper is still worthy of acceptance, but am marking "less certain" because of this factor.



**Award:**

No

---

### Decision · Program_Chairs · 2022-09-14

Accept